# *Gcorn fungi*: A Web Tool for Detecting Biases between Gene Evolution and Speciation in Fungi

**DOI:** 10.3390/jof7110959

**Published:** 2021-11-12

**Authors:** Taiga Kawachi, Yuta Inuki, Yoshiyuki Ogata

**Affiliations:** Graduate School of Life and Environmental Sciences, Osaka Prefecture University, Sakai, Osaka 599-8531, Japan; sac02029@edu.osakafu-u.ac.jp (T.K.); sac02010@edu.osakafu-u.ac.jp (Y.I.)

**Keywords:** database, fungi, evolution, gene homology, network analysis, ortholog, phylogenetic tree, speciation

## Abstract

(1) Background: Fungi contain several millions of species, and the diversification of fungal genes has been achieved by speciation, gene duplication, and horizontal gene transfer. Although several databases provide information on orthologous and paralogous events, these databases show no information on biases between gene mutation and speciation. Here, we designed the *Gcorn fungi* database to better understand such biases. (2) Methods: Amino acid sequences of fungal genes in 249 species, which contain 2,345,743 sequences, were used for this database. Homologous genes were grouped at various thresholds of the homology index, which was based on the percentages of gene mutations. By grouping genes that showed highly similar homology indices to each other, we showed functional and evolutionary traits in the phylogenetic tree depicted for the gene of interest. (3) Results: *Gcorn fungi* provides well-summarized information on the evolution of a gene lineage and on the biases between gene evolution and speciation, which are quantitatively identified by the Robinson–Foulds metric. The database helps users visualize these traits using various depictions. (4) Conclusions: *Gcorn fungi* is an open access database that provides a variety of information with which to understand gene function and evolution.

## 1. Introduction

Fungi are one of the most diverse groups of organisms, are essential to ecosystems, and are closely related to the soil carbon cycle, plant nutrition, and pathology [1]. The estimated number of fungal species on the earth ranges from 1.5 million to 10 million [2]. Although it is difficult to know the exact number of fungal species, there is no doubt that they represent a high degree of eukaryote diversity.

The diversification of genes is, in general, due to speciation, gene duplication, and horizontal gene transfer. There are various types of speciation (e.g., allopatric and sympatric) [3]. In any type, speciation is driven by natural selection over a long period of time. In contrast, gene duplication events increase the copy numbers of genes, and then mutations occur in the duplicates, which promotes biodiversity [4]. Although these are based on vertical gene transfer, diversification is also achieved by horizontal gene transfer [5,6,7].

Tracking such diversification events (speciation, gene duplication, and horizontal gene transfer) helps uncover evolutionary pathways. Genes of organisms branched from a common ancestor by a speciation event are named “ortholog” [8]. An analysis of orthologs is one current way to estimate speciation. Genes that are advantageous to the survival of an organism, fortuitously duplicated, and then adapted to various biological functions are named “paralogs”. Although paralogs initially shared the same function with each other, they are more likely to have evolved to obtain different functions according to purifying selection [9]. Therefore, tracking orthologous and paralogous events allows for the functional annotation of ancient genes as well as the timing of speciation.

Molecular phylogenetics is based on sequences of genes that are mainly included in the genomes of mitochondria and plastids. Although nuclear genes can be, in general, used for reconstructing phylogenetic trees, these trees are different from those based on the genomes of mitochondria and plastids [10]. In other words, there are biases between a phylogenetic tree based on particular genes and the tree of speciation. When reconstructing a phylogenetic tree, such biases create errors in the reconstruction. In contrast, the biases help to understand the functions of ancient genes by tracing a fluctuation in the frequency of gene mutations. Our previous reports [11] discussed the roles of biases in inferring gene function.

For tracking orthologous and paralogous events for a gene of interest, several databases that contain information on homologs of fungi, such as Ensemble Fungi [12], OrthoDB [13], AYbRAH [14], and FungiDB [15], which are periodically updated, are useful. Ensemble Fungi is an integrated database that contains all kinds of information on phylogenetics. The database has a function to depict a phylogenetic tree that contains a gene of interest and its homologous genes. The OrthoDB database contains not only information on fungal genes but also information on genes from various species of other eukaryotes and prokaryotes. Furthermore, the number of genes and species in this database is quite large, and its information is well organized. The AYbRAH database contains information on yeast genes. By reconstructing the ancestors of homologs, this database has achieved more accurate gene grouping than the other databases. The FungiDB database contains information on genes of fungi and oomycetes, and is a free, online resource for data mining and functional genomics. However, these databases show no information on biases between gene mutation and speciation.

Here, we have developed the *Gcorn fungi* database to evaluate the functional and evolutionary properties of fungal genes from sequence homology and to better understand the biases between gene mutation and speciation. We used the BLASTP program [16] for sequence homology analysis between fungal genes obtained from the NCBI Reference Sequence database (RefSeq) [17]. *Gcorn fungi* is designed to provide information on the dynamics of homologous gene groups at consecutive mutation points by grouping fungal genes with successive thresholds of homology indices (*HI*), which we previously introduced in [11]. The *Gcorn* project aims to provide users with useful information on the evolutionary dynamics of genes that transmit in both vertical and horizontal manners, which will lead to a more detailed understanding of the evolution of genes and organisms.

## 2. Materials and Methods

### 2.1. Schema

The majority of the *Gcorn fungi* database was constructed in a manner similar to the *Gcorn plant* database [11]; therefore, their basic parts are described in the previous paper. In this section, only the newly introduced items are addressed.

Amino acid sequences of fungal genes were obtained from the RefSeq database [17], and BLASTP [16] analysis was performed between all pairs of genes.

### 2.2. Data Source

Files were obtained from RefSeq (https://ftp.ncbi.nlm.nih.gov/refseq/release/fungi/, accessed on 22 November 2017). These files contained 2,351,993 sequences from 507 fungi (mainly species). However, there was variation in the numbers of genes contained in individual fungi. Because the *Gcorn* project aims to compare homologous genes at the genome level, 249 fungi, which contain 1000 or more genes, were selected for *Gcorn fungi* (Appendix A). In the selected fungi, there were 2,345,743 sequences (99.7% of the original sequences).

### 2.3. Implementation

The BLASTP analyses were executed using the following codes: “blastp -outfmt ‘6 std qlen slen’ -max_target_seqs 5000”. The “qlen” and “slen” options were used for calculating a homology index in a single line of a result file of the BLASTP search. Although the “max_target_seqs” option should have been 10,000 or more, a value of 5000 was used to alleviate the error rate of the blast search, based on our preliminary analyses. Therefore, a homologous group that contains 5000 or more genes can have incorrect membership.

The Robinson–Foulds (RF) metric [18] was used to calculate distances between pairs of phylogenetic trees.

### 2.4. Detection of Homologous Gene Groups

In total, 1,861,474 gene groups of fungi were detected, and their types of homology (i.e., paralogous or orthologous) were determined in the database as follows.

A lineage of a gene of interest is exemplified for the determination (Figure 1). In the lineage figure, three types of line charts are contained, i.e., red (the number of genes), blue (that of species or strains), and green (that of families) lines. Each dot in the figure represents a gene group. The horizontal and vertical axes represent the sequence similarity (0 to 1) in the group, which is along evolutionary time, and the numbers of genes, species or strains, and families of the group. Here, we focus on a pair of adjacent dots in red and blue. When the number of genes decreases rightwards, a homologous event hypothetically occurred in the term of the pair. When the number of species (or strains) decreases or is unchanged in the pair, an orthologous or paralogous event hypothetically occurs, respectively.

These groups are identified as the “GFH” groups in *Gcorn fungi*, which represent “Gcorn Fungi Homology” groups, and are not necessarily coincident with orthologous gene groups in the other phylogenetic trees. In other words, The GFH groups are based on the sequence similarity calculated by the BLASTP analyses for the present database.

### 2.5. Construction of a Phylogenetic Tree for Species (Taxonomy Tree)

A taxonomy tree that contains 249 fungi was constructed based on phylogenetic relationships obtained from the NCBI Taxonomy database [19] and named “Taxonomy tree”. The relationships are listed in the “nodes.dmp” file, which stores the “parent-child” relationship of nodes that represent taxa. When branches in the relationships were trifurcated or more, such branching was alleviated using information from other studies that show detailed relationships on the branches [20,21,22,23,24,25,26,27,28,29,30,31,32,33,34,35,36,37,38,39,40,41]. However, there still remained some branches that were trifurcated or more after the alleviation.

### 2.6. Construction of a Phylogenetic Tree for Genes (Orthology Tree)

The *Gcorn fungi* database provides a phylogenetic tree based on only orthologous divergence by using a gene for a species that is the most homologous to a target gene, named “Orthology tree”. This phylogenetic tree is similar to the Taxonomy tree, although there are some branches dissimilar to speciation. The discrepancy between the Taxonomy tree and the Orthology tree can directly represent that between gene evolution and speciation, leading to a deeper consideration of the evolution of genes and organisms.

## 3. Results

### 3.1. User Interface

Figure 2 shows the configuration of the *Gcorn fungi* website. Gcorn fungi is available without user registration at http://www.plant.osakafu-u.ac.jp/~kagiana/gcorn/f/ (accessed on 22 November 2017). A user can search for a gene of interest in its portal site to access pages for information on the gene’s homology (Homology page, hereafter) and orthology (Orthology page, hereafter). A Homology page contains a phylogenetic tree of a gene group that contains the gene and line charts based on information on a lineage of gene groups that contain the gene. An Orthology page contains a species–species network based on the ratio of homologous genes shared between the species studied, a Taxonomy tree, and an Orthology tree, based on the taxonomy database and gene orthology.

Although the search method from the portal site is similar to that of *Gcorn plant*, an “advanced search” function was additionally introduced (Figure 3), which makes it possible to search for a gene from multiple taxa and to find functional annotations of multiple genes.

Instead of selecting a single species, *Gcorn fungi* can accept all species studied in fungi. In the second step for “Gene search” on the portal website, a user can select “all fungi” at the bottom of the selector.

In the display subsequent to the portal site (Appendix A), the user is required only to select a gene of interest from a table of candidates and then click on its “Homology” or “Orthology” button to show its “Homology page” or “Orthology page.”

### 3.2. Homology Page

The Homology page contains two sections: one is a phylogenetic tree of a gene group that contains a gene of interest, and the other is a set of line charts that show a history in the homologous events associated with gene groups that contain the gene. These sections are similar to those in *Gcorn plant* [11].

### 3.3. Orthology Page

The Orthology page provides information on the relationship between gene evolution and speciation. This webpage contains a species–species network based on gene homology [11] (Figure 4 and Appendix A) and two phylogenetic trees based on taxonomy (Appendix A) and gene orthology (Appendix A), respectively.

In these networks, nodes represent species (organisms), and are connected to others based on the *CI* index as follows:$$CI=\frac{2N_{S}}{\left(N_{C}+N_{D}\right)}$$
where *N_S_* represents the number of genes homologous to each other and *N_C_* and *N_D_* represent the numbers of genes contained in their genomes, respectively. This index is also coincident with the F-measure. Red dots represent species that share homologs to the gene of interest at the threshold of the HI, which is changeable by a user. In this figure, the networks are depicted at a threshold of 0.6. Figures with different thresholds are in Appendix A.

In the Taxonomy tree, species (or strains) that contain orthologs to the gene are displayed in red as scientific names. The RF metric and its percentile, which are shown together to show the discrepancy between the two phylogenetic trees, quantify the degree of deviation between the two trees. The RF metric tends to be higher, with a greater number of species contained in the phylogenetic tree. To solve this problem, a percentile is a comparable metric that is calculated for each number of species that make up the phylogenetic tree of the orthologous genes. Specifically, a high percentile represents high similarity between the Orthology tree for a gene and the Taxonomy tree, which contains species that are the same as those in the Orthology tree.

### 3.4. Other Downloadable Datasets

On the Orthology page, by clicking the “Ortholog table” and “Ortholog FASTA” buttons, the database shows a table and a multi-FASTA formatted text document of genes orthologous to a gene of interest, respectively. This FASTA file is available for multiple alignment tools such as Mega [42] and ClustalX [43].

## 4. Discussion

The *Gcorn fungi* database provides not only information on relatively new homologous events related to a gene of interest, but also well-summarized information on the evolution of a gene lineage, which is depicted by classifying homologous gene groups, and on the bias between gene evolution and speciation, which is quantitatively identified by the RF metric [18], which shows the distance between a pair of phylogenetic trees. In other databases that focus on gene orthology, such as Ensemble Fungi [12], OrthoDB [13], and FungiDB [15], there are no functions to show such biases because these databases aim to provide information based on a precise phylogenetic tree.

However, there are biases that complicate gene evolution and speciation [9]. Such biases can be due to various events, such as convergent evolution and horizontal propagation, and thus these biases are useful to estimate evolutionary events along with speciation. The goal of the *Gcorn* project is to describe gene evolution and function based on biases between gene mutation and speciation. In detecting such biases, a comprehensive search of homologous gene groups is required.

This comprehensive tracking enables the consideration of the complete history of gene evolution based on detectable mutations. In particular, it is useful to accurately detect homologous events that have occurred in the gene of interest. The comprehensive tracking of orthologous events provides hints for estimating the exact timing of speciation, even if such speciation is still controversial. By comprehensively tracing a gene lineage, the exact timing of genome or gene duplication can be estimated. When paralogous events have co-occurred in many genes of a target species, it is evidence that these events are due to the whole or part of genome-level duplication; when paralogous events occur only in a particular gene or gene region, it is evidence that these events are due to localized duplication in the genome, such as due to transposition.

An approach using the lineage of a gene of interest can reveal the existence of horizontal gene transfer. Although the present version of *Gcorn fungi* has no function to resolve horizontal gene transfer, a chart of the lineage will contain information on horizontal gene transfer. The majority of lineage charts contain almost all species studied; in contrast, homologous gene groups are, in some charts, untraceable, in order to contain all fungal species. This means that these charts have the potential to reveal horizontal gene transfer.

There is, to our knowledge, no database in the field of fungal research that provides a function to compare phylogenetic trees of taxonomy and orthology, and that can be used for discussing biases between gene evolution and speciation. The *Gcorn fungi* database presents critical information to shed light on the mysteries of gene evolution, based on their evolutionary biases related to taxonomy and orthology trees.

### 4.1. Case Study

As an example, to retrieve information obtained from *Gcorn fungi*, we selected a gene of interest: NP_015154.1, which functions as “NADPH dehydrogenase” in *Saccharomyces cerevisiae*. Since NADP acts as a coenzyme for amino acid dehydrogenases in various organisms, the amino acid sequence of the gene may be highly conserved. Therefore, it is expected that this gene can be used for DNA barcoding, which is discussed in the remainder of this section.

Figure 5 shows a phylogenetic tree of a gene group that contains the gene of interest, and the tree is useful to grasp conservative traits of and homologous events in the group. Since all of the branches in the tree are connected by high HIs of 0.7 to 0.8, the homology of these genes is reliable. Therefore, the functions of these genes will be the same as or similar to each other. In fact, these genes, except for genes with unknown functions, all have the same functions or domains, according to their metadata. In addition, mutation events can be observed in the tree. Branches within a single strain of a species, such as those between XP_003676072.1, XP_003678492.1, and XP_003675582.1 in *Naumovozyma castellii*, indicate paralogous events. Of note, these genes are highly conserved in the species, and thus the event is possibly beneficial for understanding the evolutionary traits that aid in this species’ survival, unless they have recently been duplicated. Moreover, branches between other species, such as a branch between NP_012049.1 and XP_018221956.1, indicate orthologous events. There are several orthologous events in the tree, and the orthologs are conserved in many species. Based on the high conservation in these genes, the gene of interest could be used for DNA barcoding.

Figure 1 shows an evolutionary lineage chart of the gene of interest with a set of line charts that represent the numbers of genes, species, and families. The chart is useful to investigate a wide evolutionary range of conservative traits and to detect homologous (i.e., orthologous and paralogous) events of the gene at a high resolution. In Figure 1, each dot represents a homologous gene group, and the lineage of the gene is retrospective leftward from the right end (i.e., evolutionary time progresses rightward). This lineage chart is a summary of a phylogenetic tree that contains the gene and the other genes homologous to it, and is useful for focusing on a homologous gene group in the lineage. Leftward from the right end of the lineage, at an HI of 0.814, a homologous gene group contains 15 genes, 10 species, and a family; thus, these homologous genes are highly conserved in Saccharomycetaceae. At HIs of 0.814 to 0.551, the homologous genes are conserved within a group that contains particular taxa (i.e., Saccharomycetaceae, Phaffomycetaceae, and Debaryomycetaceae). The genes shared with these families show reasonably high homology indices. When the HI gets lower, the group is merged with a larger (over a thousand) group of genes, showing that many fungal species possess a variety of homologous traits to the gene of interest, even though the gene homology is quite limited, such as in a single functional domain.

Figure 1 also classifies the homologous events of the gene of interest into orthologous and paralogous events. An orthologous event is detectable when both lines that represent the numbers of species (species line, hereafter) and gene (gene line, hereafter) increase leftward, and a paralogous event is detectable when a species line is parallel to the horizontal axis and the gene line increases. For example, in Figure 1, the homologous event that occurred between HIs of 0.551 and 0.544 is presumably an orthologous event, because both gene and species lines rapidly increase. The homologous event that occurred between 0.750 and 0.650 is presumably a paralogous event, because only the gene line increases. A total of 193 homologous events are detectable from the lineage chart, which contains major (many genes divided into two main groups) and minor (a few genes were separated from the main group) events. This means that many homologous events can be detected and classified by using a successive threshold of the HI, and they contain 10 or fewer major events, based on the main taxa.

The *Gcorn fungi* database provides a network between fungal species based on the ratio of genes shared with other fungi studied, based on gene orthology (Figure 4 and Appendix A). The coloration of nodes in this network is based on a threshold of HI that is adjustable by the user (by default, the threshold is 0.8); i.e., a red or grey node represents species with or without a gene orthologous to a gene of interest. In the networks shown in Figure 4 and Appendix A, orthologs are emphasized in red at different thresholds of 0.9 (high in Appendix A), 0.6 (moderately high in Figure 4 and Appendix A), and 0.3 (low in Appendix A). At the high threshold, no genes orthologous to the gene of interest were detected. At the moderately high threshold, orthologous genes were detected in the same family (i.e., Saccharomycetaceae). At the low threshold, such genes were detected in most species over families. Although many orthologs are detected at a low threshold, the detected orthologs do not necessarily share the same function with each other. The functionality may be lost or changed throughout their evolution. Their amino acid sequences may be similar due to limited domain regions, such as a nuclear localization signal. In some cases, the gene sequences may become similar to each other as a result of convergent evolution; i.e., not based on gene orthology. By adjusting the threshold, a user can draw a network reliable for detecting orthologs. Furthermore, by gradually changing the threshold values, it is possible to directly visualize how genes homologous to the gene of interest have evolutionarily transitioned. The threshold used in Appendix A (i.e., HI of 0.6) is reasonable for estimating genes orthologous to the gene of interest. Based on this estimation, orthologous genes to the gene of interest are highly conserved within Saccharomycetaceae.

Appendix A show conservative traits of the gene of interest across species and biases that occurred during gene evolution and speciation. Appendix A shows that many species have preserved genes orthologous to the gene of interest. The RF metric (269) and its tree percentile (75.6%) between the trees are reliable in showing similar branching, indicating that gene evolution and the speciation of genes orthologous to the gene of interest are relatively close to each other. However, there were some different branches between the trees. Focusing on *Aspergillus* (Appendix A), the branches between *A. nidulans* and *A. glaucus* are inconsistent with each other. This indicates that the gene of interest has a low potential to contribute to the presumed timing of speciation in *Aspergillus*. This kind of different branching is detectable in most fungal genes. Based on these features, the Taxonomy and Orthology trees are useful not only to detect differences between speciation and gene evolution, but they can also provide detailed information on the timing of speciation. By focusing on coincidence in the branches between the Taxonomy and Orthology trees, information on the branching based on a higher resolution than that in the Taxonomy tree (i.e., at a lower resolution) is available. For example, focusing on *Kwoniella* (highlighted in Appendix A), species of the genus are classified into the same branch in the Taxonomy tree, and, in contrast, they are further branched at a high resolution in the Orthology tree. Because of such fine branching, a user can get a hint of the refined phylogenetic estimation from the Orthology tree.

In this case study, we found much information on the homology of the genes of interest (NP_015154.1) obtained from *Gcorn fungi*. The conservative traits of the gene are summarized as follows:Homologous genes are detected throughout fungi.The gene sequence is well conserved in Saccharomycetaceae.Although there are some biases between the gene evolution and speciation of the gene of interest, evolution is generally consistent with speciation.The gene of interest has the potential to serve as a reference gene for DNA barcoding based on the above consistency.

### 4.2. Future Development

In the *Gcorn* project, analyses of protozoa, vertebrates, and invertebrates have been completed and their databases are under construction. *Gcorn fungi* is planned to be updated as soon as these databases are constructed. In the next update, we will provide a function with which to visualize the discrepancy between the Taxonomy and Orthology trees for better understanding of the biases in gene evolution, as well as adding genes and species that are newly released in the RefSeq database.

Furthermore, in the future we plan for the *Gcorn* project to be able to comprehensively retrieve horizontal gene transfer information. In general, calculations for sequence homology in gene pairs between different kingdoms have time and computational costs. The majority of these calculations provide low values for sequence similarity, and thus result in useless information. An algorithm that uses only gene pairs valid for the construction of the database will enable us to detect horizontal gene transfer between kingdoms and, moreover, to use a database for prokaryotic genes, which are tens of times more common than eukaryotic genes in the RefSeq database.

## 5. Conclusions

*Gcorn fungi* provides information for clarifying gene evolution based on biases between gene mutation and speciation in 249 fungal species. This information will be of great help to users for understanding the dynamics of gene evolution. The database is freely available for academic use at http://www.plant.osakafu-u.ac.jp/~kagiana/gcorn/f/ (accessed on 22 November 2017).

## Figures and Tables

**Figure 1 jof-07-00959-f001:**
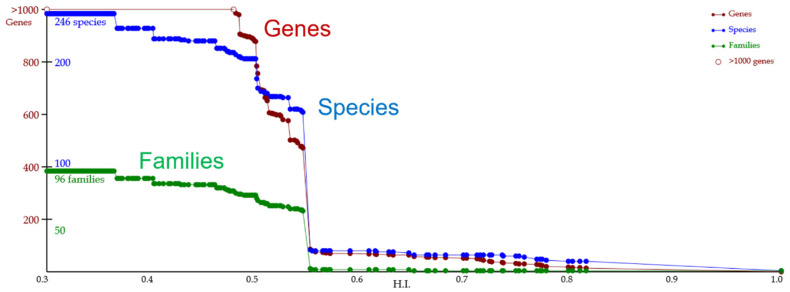
Line charts for a lineage of gene groups for the *S. cerevisiae* gene NP_015154.1. This diagram is depicted on the Homology page. The horizontal axis is the *HI*. Each dot represents a homologous gene group, which is hypothetically equivalent to a gene in an ancient organism. Red, blue, and green lines represent the numbers of genes (sequences), species, and families contained in individual gene groups, respectively. These numbers represent one of the groups’ descendants in extant organisms. Tracking groups of homologous genes is based on the *HI*. Evolution time goes from the left edge to the right edge.

**Figure 2 jof-07-00959-f002:**
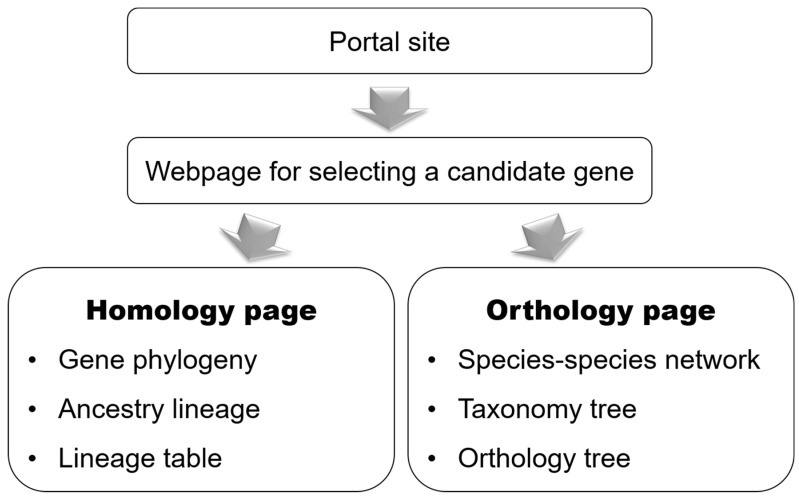
Flowchart of data retrieval for *Gcorn fungi*. A user inputs information on a gene of interest on the portal site (Figure 3). For a detailed description for each page, see each figure for the page.

**Figure 3 jof-07-00959-f003:**
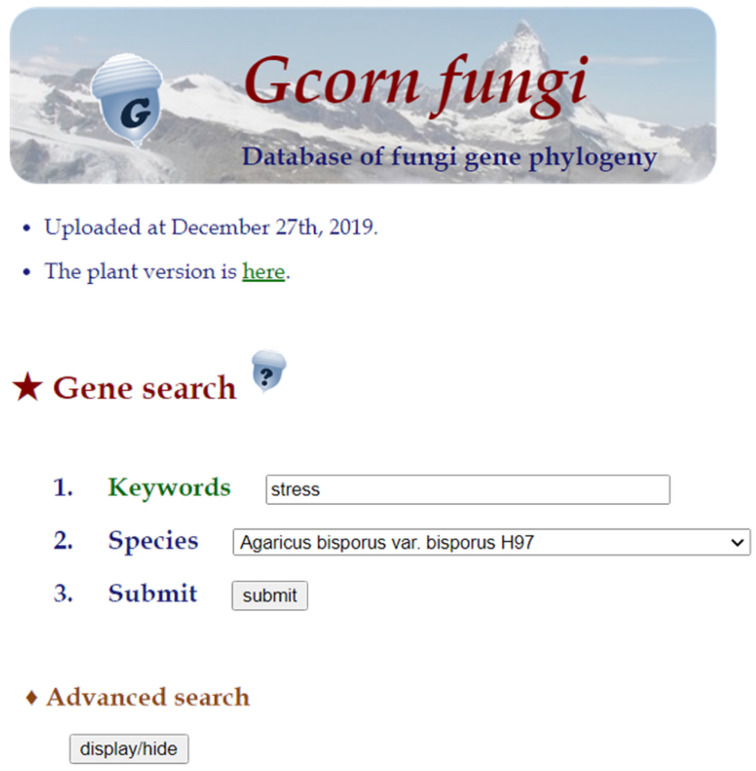
The portal site of *Gcorn fungi*. In the portal site, there are just three steps: (1) in the “Keywords” item, input a gene ID or words related to gene function, such as “stress”, (2) in the “Species” item, select a species from the pull-down menu, and (3) click the submit button. In “advanced search”, the database provides four additional functions: (1) to search for information on a single gene from multiple species or taxa and to retrieve information on (2) gene annotations, (3) orthologs, and (4) Gene Ontology terms of multiple genes.

**Figure 4 jof-07-00959-f004:**
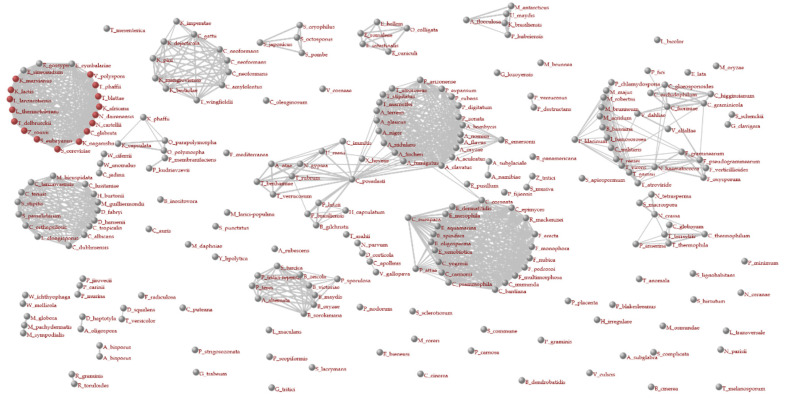
Species–species networks for the *Saccharomyces cerevisiae* gene NP_015154.1.

**Figure 5 jof-07-00959-f005:**
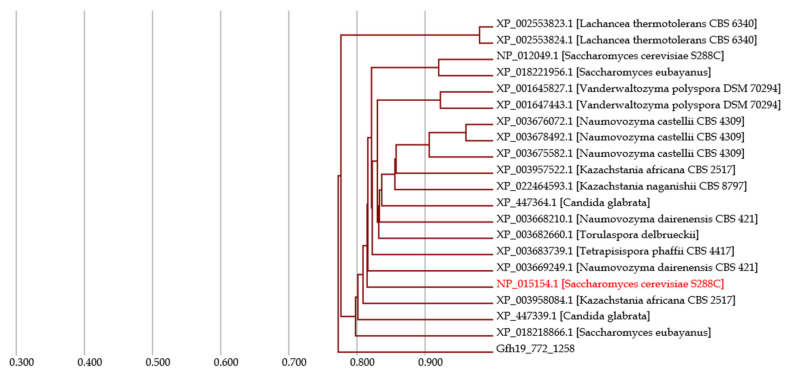
Phylogenetic tree for the *S. cerevisiae* gene NP_015154.1. On the Homology page, the phylogenetic tree for the gene group to which the gene of interest belongs is depicted. The horizontal axis represents the HI. The gene in red is the gene of interest, and the others are genes that represent its orthologs based on HIs. Evolutionary time goes from the left edge to the right edge.

## Data Availability

The datasets used for constructing *Gcorn fungi* were all obtained from the RefSeq database of the NCBI. No original datasets were used for the construction.

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
