# Peer review of "Gcorn fungi: A Web Tool for Detecting Biases between Gene Evolution and Speciation in Fungi"

_jof, 2021, doi:10.3390/jof7110959_

Round 1
Reviewer 1 Report
This manuscript "Gcorn fungi: A web tool for detecting biases between gene evolution and speciation in fungi" is an interesting piece of work worth publishing in JOF but it needs some revisions before it is published. The comments and suggestions are annotated in the manuscript. This can be accepted after the suggested revisions are properly done.

Reviewer 2 Report
It is a clear description of the database and web tool named “Gcorn fungi” developed by the authors. It was constructed similarly to their “Gcorn plant” database published earlier. I checked how it works; it is a user friendly and useful platform. I suggest publishing the manuscript. One suggestion, not to the paper, but to the database: it would be nice to complete it with the data available in 1000 Fungal Genomes project (MycoCosm fungal genomics resource), especially with the data of early branching fungi (Grigoriev et al, 2014. Nucleic Acids Res. 42, D699eD704).
Minor points:
References: Use always italic letters for the species and genus names.
Ref. 21: instead of “exophiala” use “Exophiala”
line 374: Use “plan” instead of “plant”.
Author Response
Thank you for the useful comments. According to the comments, we have corrected the names of genera and species in References, and have corrected the spelling in line 374.